# Evaluation of the Sheet Resistance of Inkjet-Printed Ag-Layers on Flexible, Uncoated Paper Substrates Using Van-der-Pauw’s Method

**DOI:** 10.3390/s20082398

**Published:** 2020-04-23

**Authors:** Johanna Zikulnig, Ali Roshanghias, Lukas Rauter, Christina Hirschl

**Affiliations:** Silicon Austria Labs GmbH, Inffeldgasse 33, 8010 Graz, Austria; ali.roshanghias@silicon-austria.com (A.R.); lukas.rauter@silicon-austria.com (L.R.); christina.hirschl@silicon-austria.com (C.H.)

**Keywords:** printed electronics, inkjet printing, paper substrate, Van-der-Pauw, sheet resistance, additive manufacturing

## Abstract

With the growing significance of printed sensors on the electronics market, new demands on quality and reproducibility have arisen. While most printing processes on standard substrates (e.g., Polyethylene terephthalate (PET)) are well-defined, the printing on substrates with rather porous, fibrous and rough surfaces (e.g., uncoated paper) contains new challenges. Especially in the case of inkjet-printing and other deposition techniques that require low-viscous nanoparticle inks the solvents and deposition materials might be absorbed, inhibiting the formation of homogeneous conductive layers. As part of this work, the sheet resistance of sintered inkjet-printed conductive silver (Ag-) nanoparticle cross structures on two different, commercially available, uncoated paper substrates using Van-der-Pauw’s method is evaluated. The results are compared to the conductivity of well-studied, white heat stabilised and treated PET foil. While the sheet resistance on PET substrate is highly reproducible and the variations are solely process-dependent, the sheet resistance on uncoated paper depends more on the substrate properties themselves. The results indicate that the achievable conductivity as well as the reproducibility decrease with increasing substrate porosity and fibrousness.

## 1. Introduction

In recent years, printed electronics have found their way into commercial applications with the aim to make electronic applications cheaper, flexible and integrable in all sorts of materials and wearables [1,2,3]. The growing significance of printed electronics is mainly due to their high potential for cost-efficient mass manufacturing and a high degree of customisation. In addition, as an additive method, printing is considered to have less environmental impact than classical electronics manufacturing [4,5]. Considering sustainability, the applicability of paper as flexible substrate in printed electronics has been attracting continuing interest for several years [6,7,8,9,10], since paper is a low-cost, easily available and biologically degradable material. A recent simulation study also indicates that, in the case of landfilling paper-based electronics, less potentially harmful metal ink particles are released into the environment than when using polymer-based substrates [11]. For the application in printed sensors, the characteristic porosity and surface roughness of paper can even be exploited. An example would be the capacitive sensing of humidity, where the porous paper serves as a dielectric material. When the paper absorbs humidity from its environment, the dielectric constant of the capacitor alters accordingly, and those changes can be measured [12,13,14,15]. Additionally, wireless readout-options for such sensors have frequently been reported [16,17,18] paving the way for truly low-cost, sustainable and smart packaging solutions of the future [19,20]. At the same time, the paper’s porosity as well as surface inhomogeneity and higher roughness compared to widely used polymer-based substrates create new challenges for the printing process, especially for printing methods that require low-viscous inks, such as inkjet-printing [6,21,22]. To reduce the influence of these inherent disadvantages, coating is widely applied as it has proven to be highly effective to gain control over the ink–substrate interactions [23,24,25]. Depending on the intended use, different coating layers can be applied, such as top coatings commonly consisting of Aluminium oxide (Al_2_O_3_) [26], kaolin [25,27], or Calcium carbonate (CaCO_3_) [7], together with polymer-based binders, such as polyvinyl alcohol or polyurethane [25]. Additionally, barrier layers or so-called “pre-coatings” might be required to separate the top coating from the paper substructure. For this purpose, resins [28], polyethylene [26] or other polymer-based materials [25] are used. Despite the obvious advantages of coating, it is an additional time- and resource-consuming processing step that increases the resulting material price. Furthermore, although there are approaches towards the development of more ecological (e.g., nanocellulose-based [29]) coatings, commonly applied polymer-based coatings cannot be considered as environmentally friendly, revoking the benefits of paper as a substrate for sustainable electronic development. Consequently, there is a desire to establish stable and reproducible printing procedures on uncoated paper substrates.

Therefore, it needs to be considered that for most printed sensors, e.g., for resistive sensing applications [30,31,32,33], it is crucial to design structures with a predefined total resistance that guarantee some degree of reproducibility, as otherwise each sensor would have to be calibrated individually. However, inkjet-printed conductive layers on porous substrates (e.g., paper) have no homogeneous surface, as well as a varying thickness of a few micrometres, which is little related to its planar dimensions [34]. As the determination of the volume resistivity is not trivial in this case, the sheet resistance is measured. The sheet resistance represents the electrical resistance of a two-dimensional, extended and homogeneous square-shaped plane, and is commonly used in the semiconducting industry as part of the electrical characterisation of thin films. To highlight the nature of this resistance, it is frequently given in units of Ω□ or Ωsq [35]. For the determination of the respective sheet resistance either contact-less methods, such as the seldom used eddy-current test, or contacting methods, most commonly the four-point probing, can be employed. [21,36] One variation of the classical four-point probing is the Van-der-Pauw’s method, as described in detail in [37].

In practice, the sheet resistance (or the conductivity) of printed patterns on flexible substrates is frequently determined using two-point probing [38,39,40], which is less accurate as it does not consider the contact resistance between the specimen and the instrument [36]. In another commonly employed approach, the sheet resistance is measured using four-point-probing with spring probes [41,42]. However, the sharp tips of these probes might damage the printed layer on sensitive substrates such as paper and inhibit proper contacting. The aim of this work is to determine and evaluate the sheet resistance of inkjet-printed conductive silver (Ag-) structures on two different commercially available uncoated paper substrates using Van-der-Pauw’s method and compare it to the conductivity on white-heat-stabilised and -treated polyethylene terephthalate (PET) foil. By fabricating several samples following the same printing and curing procedure, the range of variation of the sheet resistance and the dependence on the roughness and porosity of the respective substrates is observed. In 2011, Kazani et al. [43] used a similar approach for the determination of the sheet resistance of screen-printed patterns on textile substrates. However, those textiles have a very regular fibre pattern compared to commercial copy paper, as used in our approach, and viscous pastes for screen printing do not penetrate the fibers that extensively. Öhlund et al. [26] presented a comprehensive study on the surface characteristics of commercial printing paper substrates for their application in printed electronics. Although their work is fundamental in this field, they actually made no observation of the electrical characteristics of printed structures on cheap uncoated paper substrates. Unlike the present work, they could not achieve any conductivity. Another related and fundamental work was published by Ihalainen et al. [9] in 2012. They systematically observed the printing quality and electrical properties of inkjet-printed silver and polyaniline (PANI) on different paper substrates as well as on a PET reference substrate. In contrast to the present paper, they exclusively used coated paper substrates. A very recent work by Kavčič and Karlovits [40] analyses the characteristics of three different papers as substrates for the inkjet-printing of silver-nanoparticle inks. They demonstrated the high potential of invasive plant-based paper (Japanese knotweed) for sustainable printed electronics manufacturing. However, their approaches for studying the electrical properties of the substrates under investigation are quite different than in the present work. 

## 2. Materials and Methods

For the electrical characterisation of thin films, a four-point probe measurement (or Kelvin technique) is commonly applied. As part of this widely used technique, the current source and the measurement of the corresponding voltage are decoupled from each other. As illustrated in Figure 1a, the probes are equally spaced along a measurement axis, whereat a direct current is forced from tip 1 to tip 4 while the corresponding voltage drop ∆V is measured between tip 2 and tip 3. The advantage of this approach is that the measurement results are independent of the contact resistances of the instrument’s leads, in contrast to the two-point measurement (e.g., common multimeter) [36].

However, when it comes to printed electronics, the classical setup for a four-probe measurement has its limitations, as it explicitly applies to homogeneous and extended square-shaped surface resistances. Furthermore, the tips of the probes are usually quite sharp and might penetrate and damage printed structures on sensitive substrates. Another approach would be the Van-der-Pauw’s method, which can be applied for arbitrary shaped and even anisotropic specimens [37,44]. In the semiconducting industry, a frequently applied variation of this approach is the determination of the sheet resistance by utilisation of a specific Greek cross structure, as described by Enderling et al. [45]. 

In this specific case of a Van-der-Pauw measurement, a current is forced between the points AB and BA while measuring the voltage drop between DC and CD (see Figure 1b). The corresponding resistance $R_{\mathrm{DC},\mathrm{AB}}$ can then be calculated according to Equation (1). Subsequently the resistance $R_{\mathrm{AC},\mathrm{BD}}$ is calculated similarly, as described in Equation (2). The arithmetic mean of these two values can then be used to calculate the sheet resistance $R_{□}$ according to the relationship from Equation (3). If the thickness t of the printed layer is known, the actual bulk resistivity ρ (in Ω∙cm) can be calculated using Equation (4).
(1)$$R_{\mathrm{DC},\mathrm{AB}}=\frac{V_{\mathrm{DC}}-{\;V}_{\mathrm{CD}}}{I_{\mathrm{AB}}-{\;I}_{\mathrm{BA}}},$$
(2)$$R_{\mathrm{AC},\mathrm{BD}}=\frac{V_{\mathrm{AC}}-{\;V}_{\mathrm{CA}}}{I_{\mathrm{BD}}-{\;I}_{\mathrm{DB}}},$$
(3)$$R_{□}=\frac{\pi }{\ln2}\;\cdot \;\frac{R_{\mathrm{DC},\mathrm{AB}}+R_{\mathrm{AD},\mathrm{BC}}\;}{2}\;\approx \;4.53236\;\cdot \;\frac{R_{\mathrm{DC},\mathrm{AB}}+R_{\mathrm{AC},\mathrm{BD}}}{2},$$
(4)$$\rho =R_{□}\cdot t$$

The sheet resistance of inkjet-printed silver nanoparticle ink (Sicrys 115-TM119; particle size d50 = 85 nm, d90 = 120 nm; viscosity 34 cP [46]) on two different, uncoated, commercially available paper substrates after photonic curing (PulseForge 1200) is observed. One of the paper substrates has a grammage of 120 g/m^2^ (in the following, referred to as type 4) and the other one a grammage of 87 g/m^2^ (referred to as type 7) (Mondi AG). As a reference, the sheet resistance of the same ink on a heat stabilised and treated white PET foil (Kemafoil^®^ HSPL 80 W 75, Coveme) is determined as well. The structures are printed using a PIXDRO LP50 inkjet-printing system with a Fujifilm Dimatix 10 pL print head assembly. A printing resolution of 900 dpi has been chosen, with a typical drop volume lying between 6 and 9 pL. To promote the evaporation of ink solvents during printing, the substrate table was heated to 50 °C. For each substrate, 30 symmetric samples with different line length to width ratios (1:12, 1:6 and 1:4, 10 samples each) have been prepared, as illustrated in Figure 2. The samples were manufactured from two layers of ink, which is important for the printing on uncoated paper as the first layer is highly absorbed due to the substrates’ porosity [26]. The low-viscosity ink tends to penetrate the fibres, which results in a noticeable decrease in the achievable conductivity for the single-layer structure. 

The second layer was printed directly on top of the first unsintered layer, as sintering of individual layers in multilayer printing can lead to interface structures, inhibiting the formation of a vertically homogeneous coating [47]. The printed patterns on the PET substrate are sintered by employing thermal sintering in an oven for 30 min at 130 °C, as specified by the ink supplier [46]. For the photonic curing on the paper substrates, the used sintering parameters have been obtained by carefully approaching optimum settings. The overall sintering energy for the photonic curing process on the used paper substrates resulted in 2.1 J/cm^2^. Note that it is essential to sufficiently dry the printed patterns before applying photonic curing, as otherwise the remaining solvents might reach their boiling point and may destroy the printed layers due to liquid expansion and the formation of bubbles.

The roughness of the substrates is calculated as defined in ISO 25,178 [48]. As described in Equation (4), S_a_ is the arithmetic mean of the measured absolute height Z(x,y) over a defined sampling area A. Additionally, the roughness can be characterised using S_q_ (see Equation (5)), which corresponds to the root mean square value of the measured axial height values Z(x,y).
(5)$$S_{a}=\frac{1}{A}\iint^{\;}\left|Z\left(x,y\right)\right|\mathrm{dxdy},$$
(6)$$S_{q}=\sqrt{\;\frac{1}{A}\iint^{\;}Z^{2}\left(x,y\right)\mathrm{dxdy}},$$

The actual measurement of the sheet resistance is performed using a Keithley 2700 digital multimeter. To minimise the influence of thermal electromotive forces on the low resistance measurements, an offset compensation technique is employed. During measurement, the samples are contacted with commercial and rather even crocodile clamps. All measurements, as well as the printing, curing and storage, were conducted in the same laboratory environment at room temperature (21–23 °C, 15–35%rH).

## 3. Results

### 3.1. Surface Characteristics of the Used Paper Substrates

#### 3.1.1. White Light Interferometry

The characterisation of the substrate surfaces was conducted using contactless white-light interferometry (WLI) over an area of 800 × 660 µm at an axial (z-) resolution of 1 nm. In total, four different areas on each paper type were analysed using WLI. A three-dimensional mapping of the substrate surface area of both paper substrates is illustrated in Figure 3.

Based on that, the average roughness parameters from the four measurements were calculated and reported in Table 1. The corresponding porosity values as well as the grammage for the respective paper types were provided by the manufacturer. 

#### 3.1.2. Microscopy and SEM Imaging

The numeric roughness of the used paper substrates’ surfaces is quite similar, as the results from the white-light interferometry reveal (Table 1). However, the substrates differ in their porosity as well as grammage. Furthermore, the differences in their fibrousness and the degree of surface homogeneity can be visually observed using microscopy imaging (see Figure 4). The high porosity of the type 7 paper substrate facilitates the penetration of the low-viscous ink into the fibres. This effect is particularly observable when printing single drops, as illustrated in Figure 5. Here the ink penetrates the substrate in the direction of the fibres, inhibiting the deposition of edged drops.

Microscopic images of the cross section perfectly illustrate the different nature of printed layers on non-porous PET substrate (Figure 6a) compared to the porous paper substrate (type 4, Figure 6b). While the Ag-layer on PET is quite homogeneous with an estimated thickness of 2.5 µm, the ink obviously penetrates the fibres of the paper substrate. Hence, an estimation of the layer thickness is not trivial in this case.

For a more detailed observation scanning electron microscopy (SEM) of the fully cured and sintered nanoparticles on the type 4 and type 7 substrates was conducted. The printed layer on the type 4 substrate is comparatively homogeneous with an approximate thickness of 2 µm (Figure 7). Additionally, it can be observed that the nanoparticles are well sintered. However, the images show cracks in the Ag-layer. These cracks result from the physical handling (transportation, bending, contacting during measurements, etc.) of the samples, and can be explained by the lower flexibility of the conductive pattern compared to the flexibility of the supportive substrate.

When comparing the SEM images of type 4 (Figure 7) and type 7 (Figure 8), the enormous influence of the substrate’s porosity and fibrousness on the quality of the deposited layer becomes apparent. The homogeneity of the Ag-layer on the type 7 substrate is drastically distorted by holes and massive fibres (Figure 8a). Therefore, the thickness of the layer cannot be determined. Furthermore, the low-viscous ink is highly absorbed by the porous material, as expected and described in [26].

### 3.2. Sheet Resistance Measurement Using Van-der-Pauw’s Method

The reference measurement results for the white PET substrate are listed in Table 2 and illustrated in Figure 9. It becomes clear that the sheet resistance is independent of the actual geometry of the printed pattern for the PET substrate, as the median values lie close to each other and the dispersion of the measurement values is in the range of only a few mΩ/□ (see Table 2). Still, the sheet resistance is not perfectly reproducible.

The measurement results for the paper substrates are presented in Table 3 and Table 4, as well as illustrated in Figure 10. For the paper substrates, the sheet resistance is dependent on the line to width ratio of the printed structures, respectively. The median sheet resistance value decreases with increasing cross line width and the measurement values are widely dispersed (see Table 3 and Table 4 and Figure 10).

Using Equation (4), the specific resistivity ρ of the Ag-layer on PET and the type 4 paper substrate was calculated. For the type 7 paper substrate, ρ was not calculated as the actual layer thickness could not be determined. Table 5 gives an overview on the median resistivity values for different line length to width ratios compared to the specific resistivity of bulk silver (ρ_Ag_ = 1.59 µΩ∙cm at 20 °C [49]).

## 4. Discussion

The analysis of the sheet resistance of the Ag-structures on white PET substrate revealed a quite decent, yet not perfect reproducibility of the resulting conductivity. This is due to the fact that some process parameters, such as the droplet volume (6–9 pL) and quality (formation of satellite droplets), tend to vary during printing, resulting in an altering layer thickness. One reason for the changing qualities of the droplet during processing is the substrate table heating, which increases the print head temperature as well, resulting in a decrease in the ink’s viscosity. Furthermore, the heating can promote the evaporation of solvents, which might facilitate drying of the ink at the nozzle orifice, and hence increase the risk of nozzle clogging. With the given setup, it is not possible to monitor the drop volume and quality during processing. Apart from that, the contacting of the samples might also have an influence on the measurement results, as a perfectly reproducible contacting using crocodile clamps cannot be guaranteed. The measurements were conducted in a laboratory environment where variations of the room temperature in the range of a few °C can occur. In addition, the ambient relative humidity levels are not stable either (15–35%rH). Consequently, the thermoresistive properties of silver [50] and the nanoparticles’ sensitivity to varying humidity levels [51] might also reduce the reproducibility of the measurement results.

While the variation in the sheet resistance on PET substrate is rather process dependent, the sheet resistance on uncoated paper depends more on inherent substrate properties. The influence of the sheet thickness on the resulting conductivity can be considered as negligible, as the ink is partially absorbed due to its specific porosity and the thickness of the printed layer is in the same range as the actual surface roughness. Similarly, Siegel et al. [52] observed an exponential increase in the surface resistivity with increasing surface roughness for different paper substrates. They deposited metal layers on paper using evaporation, sputter deposition or spray deposition. According to their results, the increase in resistivity is even enhanced for thinner printed layers, which they attributed to the elongated conductive pathway compared to smooth surfaces. In contrast to that, the results of the present paper emphasise the massive influence of the substrates’ porosity and fibrousness on the achievable conductivity and reproducibility. The numeric surface roughness values for type 4 and type 7 paper substrates are similar, still the resulting sheet resistance of the type 4 paper is up to ten times lower than the sheet resistance of the type 7 paper. Type 7 paper is much more porous than type 4, which leads to a high absorption of the low viscous inkjet printing ink, impeding the formation of a homogeneous layer, as illustrated in Figure 8. Despite this, all structures were printed with the same parameters, and the resulting layer thickness on the PET substrate (t = 2.5 µm) was larger than on the type 4 paper substrate (t = 2 µm) as a high amount of ink was absorbed. The determination of the layer thickness on type 7 substrate was not possible. Although the use of a highly porous and absorbing fibrous substrates led to inhomogeneous layers and lower conductivity, the ink penetration, on the other hand, increases the adhesion, and therefore the stability and durability, of the printed films [26]. Furthermore, the ink drying process tends to be accelerated, as absorption promotes the evaporation of the solvents. The fibrousness, which is illustrated in Figure 4, additionally seems to have a severe impact on the sheet resistance. The large fibres distort the printed layer and might have an insulating effect, as they cause nanoparticle separations. The orientation and size of the fibres are random, which explains the comparatively large span width of the sheet resistance around the median values, especially for the type 7 substrate (see Figure 10b). However, with increasing line width the median sheet resistance decreases for both paper substrates, as the influence of the insulating properties of the randomly oriented single fibres on the printed structures tend to decrease.

The printed Ag-layers on PET substrate were thermally sintered, while the paper samples were sintered using photonic curing. The difference in the sintering strategies is due to the fact that thermal sintering has proven to be highly stable and reproducible for the used PET substrates. In general, PET has a comparatively low glass transition temperature [53]. However, the used high-performance PET foil is heat-stabilized; more precisely, less than 0.3% heat shrinkage after 30 min at a temperature of 150 °C can be expected. In contrast to that, paper is rather sensitive to high temperatures, hence photonic curing was employed for the type 4 and type 7 substrates. The calculated bulk resistivities of the printed Ag-layers on PET and the type 4 paper substrate (see Table 5) are even lower than specified by the ink manufacturer [46], which indicates that the nanoparticles were well sintered. The specific resistivity of the Ag-layer on type 7 paper substrate was not calculated, as the layer thickness could not be determined from the measurements.

## 5. Conclusions

In this study, Van-der-Pauw’s method is utilised for the determination of the sheet resistance of cured and sintered inkjet-printed Ag-layers on two different uncoated paper substrates. The influence of the individual substrate characteristics (e.g., surface roughness, porosity, fibrousness) on the conductivity as well as the reproducibility have been thoroughly studied and the results were compared to white-heat-stabilised and -treated PET foil for printed electronics.

While the resulting sheet resistance on PET substrate appears to be mainly process-dependent and provides a decent reproducibility, the sheet resistance on the paper substrates is highly dependent on the substrate properties, such as the specific porosity and fibrousness. This is because a large amount of the low-viscous inkjet printing ink is absorbed by the substrate, which also leads to an increase in the resulting resistance. Apart from that, the absorption can be advantageous as it improves the adhesion of the printed layer to the substrate. The lowest median specific resistivity value of the sintered Ag-layer on PET substrate was 6.3 µΩ∙cm, which corresponds to four times the bulk resistivity of silver. Even on the porous type 4 paper, substrate resistivity values as low as 11.6 µΩ∙cm (7.3 × bulk) could be achieved. However, for the highly porous and fibrous type 7 paper substrate, the layer thickness and hence the specific resistivity could not be quantified.

Although the resistance values on type 4 and type 7 have proven to be less reproducible than on PET, the results do not necessarily mean that cheap and commercially available uncoated paper substrates are not qualified for the additive manufacturing of sensors. The requirements for the level of reproducibility and conductivity are highly dependent on the individual application. However, the particular properties of the material need to be considered carefully to establish a stable manufacturing process.

## Figures and Tables

**Figure 1 sensors-20-02398-f001:**
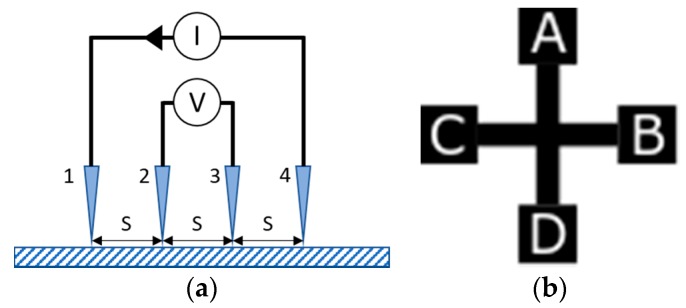
(**a**) Schematics of a four-probe measurement assembly; (**b**) Typical symmetrical cross structure for a Van-der-Pauw’s four-point-probe-measurement.

**Figure 2 sensors-20-02398-f002:**
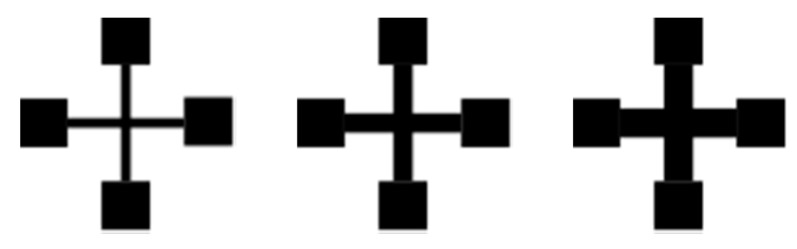
Test structures for the measurement of the sheet resistance with three different line length to width ratios.

**Figure 3 sensors-20-02398-f003:**
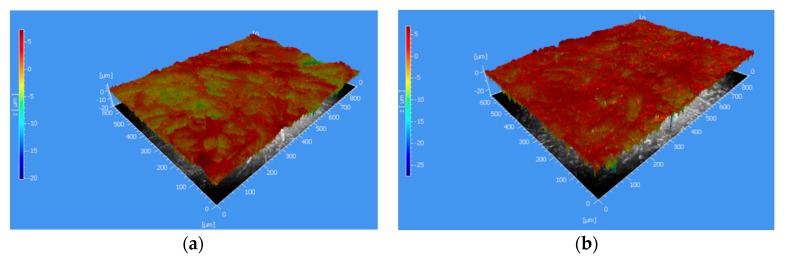
3-D white-light interferometry (WLI) mapping of the substrate surface of (**a**) type 4 and (**b**) type 7 over an area of 800 × 660 µm.

**Figure 4 sensors-20-02398-f004:**
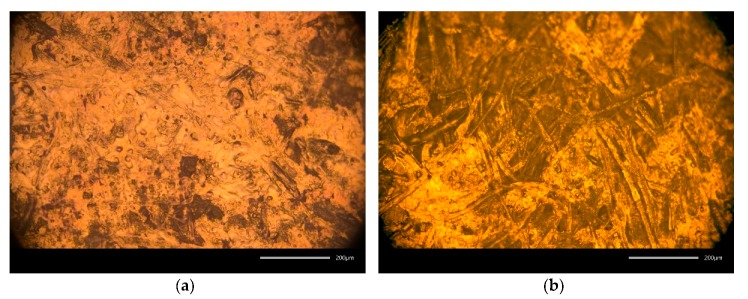
Microscopic images (20×) of the printed, cured and sintered Ag-layer on type 4 (**a**) and type 7 (**b**) paper substrate, the size bar is 200µm in length.

**Figure 5 sensors-20-02398-f005:**
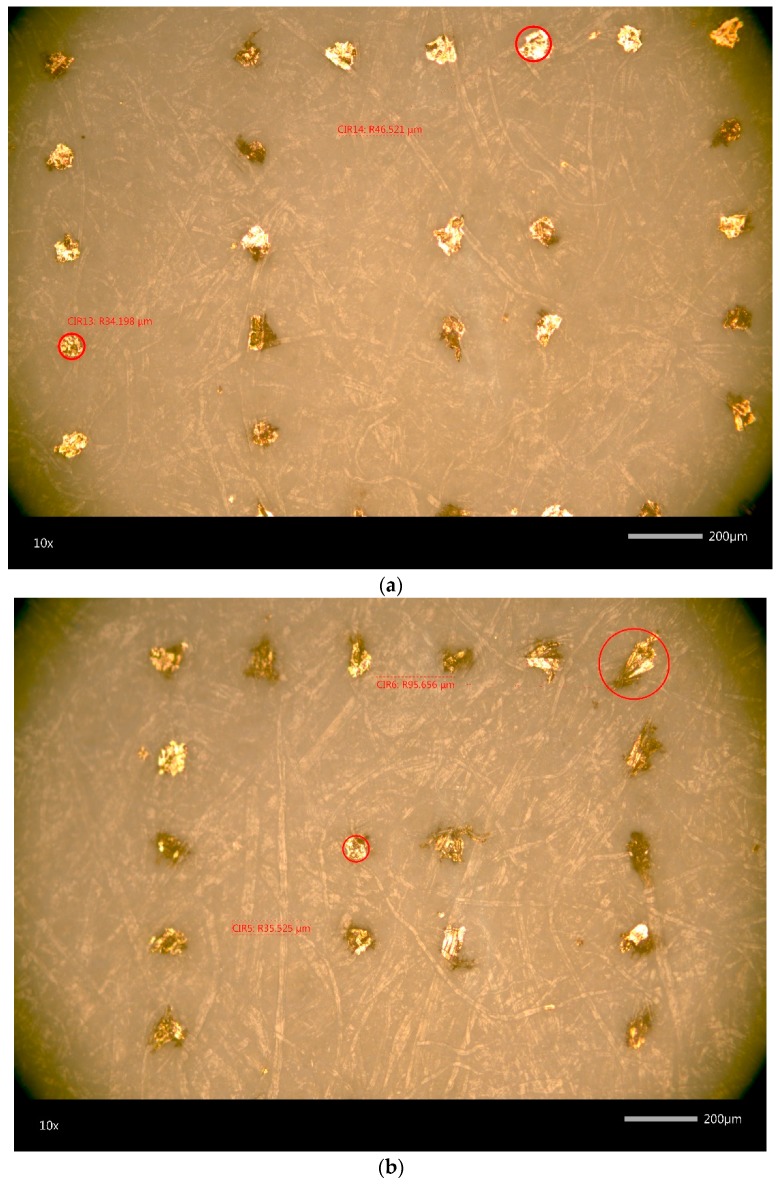
Microscopic images (10×) of printed single drops before sintering on type 4 (**a**) and type 7 (**b**) substrate, the size bar is 200µm in length.

**Figure 6 sensors-20-02398-f006:**
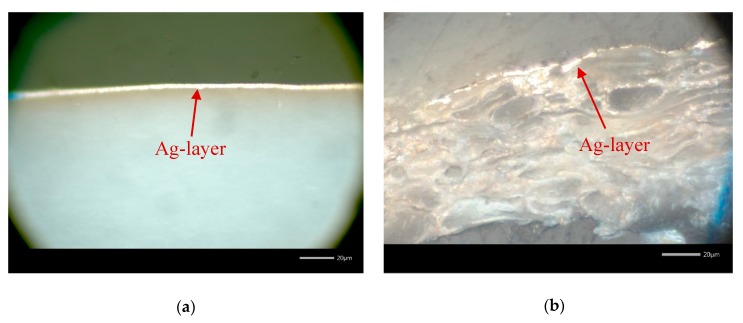
Microscopic images (50×) of the cross section of the printed, cured and sintered Ag-layer on PET (**a**) and type 4 (**b**) paper substrate; the size bar corresponds to 20 µm.

**Figure 7 sensors-20-02398-f007:**
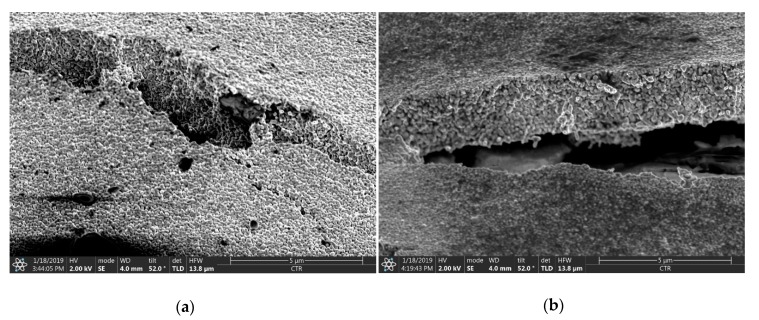
SEM images of the printed, cured and sintered Ag-layers on type 4 paper substrate; cracks in the printed layer on two different locations (**a**) and (**b**) become apparent; the size bar is 5 µm in length.

**Figure 8 sensors-20-02398-f008:**
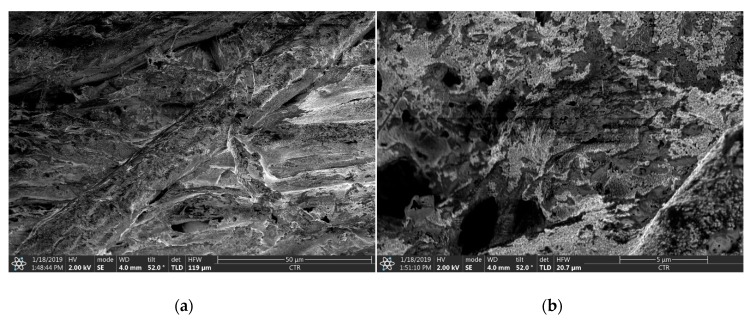
SEM images of the printed, cured and sintered Ag-layers on type 7 paper substrate; the size bars are 50 µm (**a**) and 5 µm (**b**) in length, respectively.

**Figure 9 sensors-20-02398-f009:**
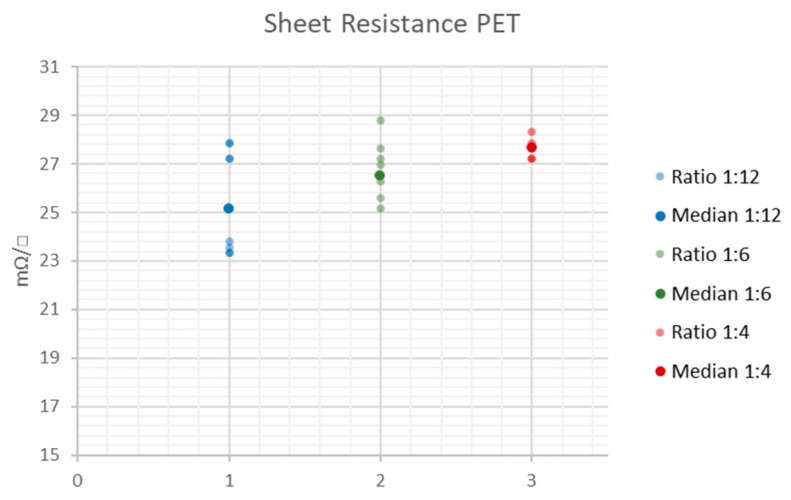
Dispersion of the sheet resistance around the median value on PET substrate.

**Figure 10 sensors-20-02398-f010:**
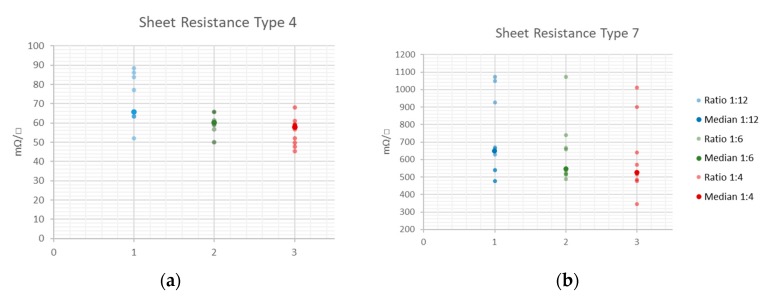
Dispersion of the sheet resistance around the median value of the printed Ag-structures on the type 4 (**a**) and type 7 (**b**) paper substrates.

**Table 1 sensors-20-02398-t001:** Characteristics of the used paper substrates.

Paper Substrate	Grammage in g/m^2^	S_a_ in µm	S_q_ in µm	Porosity in mL/min
Type 4	120	1.2	1.6	50
Type 7	87	1.4	1.9	350

**Table 2 sensors-20-02398-t002:** Statistical key data of the sheet resistance measurement values for the PET substrate.

Line to Width Ratio	1:12	1:6	1:4
Sample Size	10	10	7
Minimum (mΩ/□)	23.3	25.2	27.2
Median (mΩ/□)	25.2	26.5	27.6
Maximum (mΩ/□)	27.9	28.8	28.3

**Table 3 sensors-20-02398-t003:** Statistical key data of the sheet resistance measurement values for the type 4 substrate.

Line to Width Ratio	1:12	1:6	1:4
Sample Size	10	10	10
Minimum (mΩ/□)	52.1	49.9	45.3
Median (mΩ/□)	65.7	60.1	57.8
Maximum (mΩ/□)	88.4	65.7	68.0

**Table 4 sensors-20-02398-t004:** Statistical key data of the sheet resistance measurement values for the type 7 substrate.

Line to Width Ratio	1:12	1:6	1:4
Sample Size	10	10	10
Minimum (mΩ/□)	475.9	487.2	344.5
Median (mΩ/□)	647.0	545.0	523.5
Maximum (mΩ/□)	1071.9	1071.9	1010.7

**Table 5 sensors-20-02398-t005:** Resistivity values ρ of the Ag-layers on PET and the type 4 paper substrate for different line length to width ratios.

Substrate	ρ (1:12) in µΩ∙cm	ρ (1:6) in µΩ∙cm	ρ (1:4) in µΩ∙cm
PET (t = 2.5 µm)	6.3 (4 × bulk)	6.6 (4.2 × bulk)	6.9 (4.3 × bulk)
Type 4 (t = 2 µm)	13.1 (8.3 × bulk)	12.0 (7.6 × bulk)	11.6 (7.3 × bulk)

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
