# Peer review of "Evaluation of the Sheet Resistance of Inkjet-Printed Ag-Layers on Flexible, Uncoated Paper Substrates Using Van-der-Pauw’s Method"

_sensors, 2020, doi:10.3390/s20082398_

Round 1
Reviewer 1 Report
- It was not clear from the printing process if substrate table temperature was used. If so, which temperature and did you apply it to all the used substrates?
- The electrical measurements were performed at which humidity value, as it affects the resistivity of the printed structures?
- Why the difference in sintering methods? What is the effect of sintering on resistivity in the different substrates?
Author Response
Dear Reviewer,
Please see the attachment!
Kind regards,
Johanna Zikulnig

Reviewer 2 Report
The authors are presenting a paper called "Evaluation of the sheet resistance of inkjet-printed Ag-layers on flexible, uncoated paper substrates using Van-der-Pauw’s method".
The paper is well written and the results are clearly presented. The main problem is, that the paper does not provide anything new. Inkjet printing of silver inks on different types of substrates (including uncoated paper and cardboard) has already been studied and published. However, many such references are missing from this paper. Further, the sheet resistance measurement method brings no novelty either. If a comprehensive and systematic study about the effects of different paper substrate properties would be provided, that might be useful and bring enough new knowledge. However, now only two susbtrates are used.
Author Response

(The authors gave the same response as above.)

Reviewer 3 Report
The authors assessment the resistance of the inkjet-printed Ag-layers on two different uncoated paper substrates in this paper. To measure the resistance of the metal substrate they employed the well-known Van-der-Pauw’s method. They also show different characteristics of the printed metal substrate including conductivity, surface roughness, porosity, fibrousness, and reproducibility. The research topic of the manuscript is interesting and overall the presentation is good. In my view, the manuscript is acceptable to publish in Sensors without an additional revision.
Author Response
Dear Reviewer,
Thank you very much for taking the time to read the paper and the extremely positive feedback and kind words! We strive to publish high-quality research to the best of our knowledge and with scientific diligence. Your assessment of our work is highly motivating and encouraging for our whole research group!
Kind regards,
Johanna Zikulnig